# CONCEPT MATCHING: CLUSTERING-BASED FEDERATED CONTINUAL LEARNING

## ABSTRACT

Federated Continual Learning (FCL) has emerged as a promising paradigm combines Federated Learning (FL) and Continual Learning (CL). To achieve good model accuracy, FCL shall tackle catastrophic forgetting due to concept drift over time in CL, and overcome the potential interference among clients in FL. We propose Concept Matching (CM), an FCL framework to address these challenges. The CM framework groups the client models into model clusters, and then builds different global models to capture different concepts in FL over time. In each round, the server sends the global concept models to the clients. To avoid catastrophic forgetting, each client selects the concept model best-matching the implicit concept of the current data for fine-tuning. To avoid interference among client models with different concepts, the server clusters the models representing the same concept, aggregates the model weights in each cluster, and updates the global concept model with the cluster model of the same concept. Since the server does not know the concepts captured by the aggregated cluster models, we propose a novel server concept matching algorithm that effectively updates a global concept model with a matching cluster model. The CM framework provides flexibility to use different clustering, aggregation, and concept matching algorithms. The evaluation demonstrates that CM outperforms state-of-the-art systems and scales well with the number of clients and the model size.

## 1 INTRODUCTION

As a privacy-preserving deep learning (DL) paradigm, Federated Learning (FL) (McMahan et al., 2017) attracts significant interest. However, most of the current FL research assumes the data have been collected before training, and the data at clients do not change over the training rounds. This is not necessarily the case in many applications, especially on IoT and mobile devices where it is difficult to train with the entire dataset on-device at every round due to their resource constraints. The data not only accumulate over time, but also change their distributions. The data distribution change, also referred to as concept drift, makes DL models obsolete over time. For example, a user sleep quality prediction model trained on the data collected during their routine life will not work well when the users experience changes in their sleep patterns due to stress, illness, or travel. This effect of dynamic change in data is being actively studied by the Continual Learning (CL) community in centralized settings. However, CL research in FL settings is still in its infancy.

Federated Continual Learning (FCL) performs FL under the CL dynamic data scenarios. There are two main challenges in FCL. One, inherited from CL, is catastrophic forgetting (French, 1999). Due to concept drift, the model forgets previously learned knowledge as it learns new information over time. A concept infers a function from training examples of its inputs and outputs (Mitchell, 1997). For example, in human activity recognition (HAR) (Jiang et al., 2022), the concepts can be the subsets of activities, the locations of the activities, or the health status of the user. FCL imposes privacy constraints on the top of CL, which escalates this challenge. Even if the clients are aware of concept drift (e.g., sedentary vs. active lifestyle in HAR), they may not want to reveal it to the FL server due to privacy concerns. The second challenge is that the FL clients with different data concepts may potentially interfere with each other, because the data in FL is typically non independently or identically distributed (non-iid). The interference will sabotage the efforts of clients' training during aggregation and lead to underperforming global models. CL amplifies this interference in FL, because the union of the clients data may also be distributed differently over time.

An efficient FCL framework shall tackle these challenges to achieve good model performance. So far, no existing system has achieved this goal under realistic assumptions. While several works (Yoon et al., 2021; Casado et al., 2022; Guo et al., 2021; Zhang et al., 2023; Qi et al., 2023; Dong et al., 2022; Ma et al., 2022) have recently targeted FCL, their applicability is limited due to unrealistic assumptions (i.e., the server knows the concept drift from the clients or the classes to learn do not change over time), or the interference among the clients is not handled.

This paper proposes **Concept Matching (CM)**, the first framework for FCL to alleviate the two challenges and achieve good model performance. Intuitively, if we can separate the client models based on the data concepts, and train different models specifically to learn each concept iteratively, catastrophic forgetting and the interference among clients can be greatly diminished. This process has to be performed under the FL assumption that the server cannot access any raw data. The CM framework innovatively achieves these goals through **clustering and concept matching** in FL. At every training round, to avoid interference among the clients, the server clusters the client models representing the same concept and aggregates them. To mitigate catastrophic forgetting, different concept models are trained for each concept through concept matching which occurs differently at the server and the clients. The server concept matching is to match and update the concept model of the previous round with a cluster model. We propose a novel distance-based concept matching algorithm for the server concept matching. This algorithm matches a cluster model with a concept model close in distance, and aligns them to update the concept model in the appropriate gradient descent direction. The client concept matching is to test the concept models from the previous round on the current local data, and to select the one with the lowest loss as the best match. The CM framework does not require the clients to have any knowledge about the concepts. Furthermore, the server does not need any additional information when compared to vanilla FL (i.e., it only requires the model weights from the clients). The CM framework provides flexibility to use a variety of clustering, aggregation, and concept matching algorithms. The framework can evolve as new algorithms are proposed for different applications and models. We proved theoretically that during the iterations of gradient descent, the distance between the current model and the model from the previous round gradually diminishes. Our server concept matching algorithm applies this theorem and achieves up to 100% accuracy. Furthermore, we experimentally demonstrated the superior performance of CM over the baselines, the flexibility of using different algorithms, and its scalability with the number of clients and the model size.

## 2   CM OVERVIEW

CM is an effective learning framework for FCL, where each client encounters a stream of data with different concepts over time (Figure 1a). For example, a set of classes in image classification or an accent in speech recognition. For the photos a client takes in daily life, the concepts can be different types of the place of interest visited (e.g., museum vs. national park), or different types of meals (i.e., breakfast, lunch, dinner). The clients may or may not be aware of the concepts of the data. In FCL, it is infeasible for the clients to store all the data (e.g., limited storage on IoT devices) or the system cannot wait for all data to be accumulated before the training starts (Lomonaco and Maltoni, 2017). The system has to consume the data promptly to train one or multiple models working well for every concept. As regular FL, the server in FCL only receives and aggregates the model weights from the clients, without accessing any additional information.

The two learning scenarios (Maltoni and Lomonaco, 2019) in FCL are class-incremental and task-incremental. Class-incremental requires the system to learn different classes over time. The changes of the classes in data causes concept drift, but the system is not given any additional information to identify the concept drift. Task-incremental focuses on learning different tasks over time, such as different languages in speech recognition. This scenario can also consider the tasks to be different classes. In this case, the difference from class-incremental is that the tasks (e.g., sedentary activities vs. physical exercises in HAR) are known, and the concept drift is recognized by the model. In centralized CL, the task IDs can be used to help both training and inference. However, in FCL, the clients can utilize the task IDs, but they cannot share them with the server due to privacy concerns.

The FCL learning scenarios are negatively impacted by catastrophic forgetting due to concept drift over the training rounds and the interference among the clients with different concepts of data. CM is the first learning framework to alleviate these challenges and effectively train accurate models in

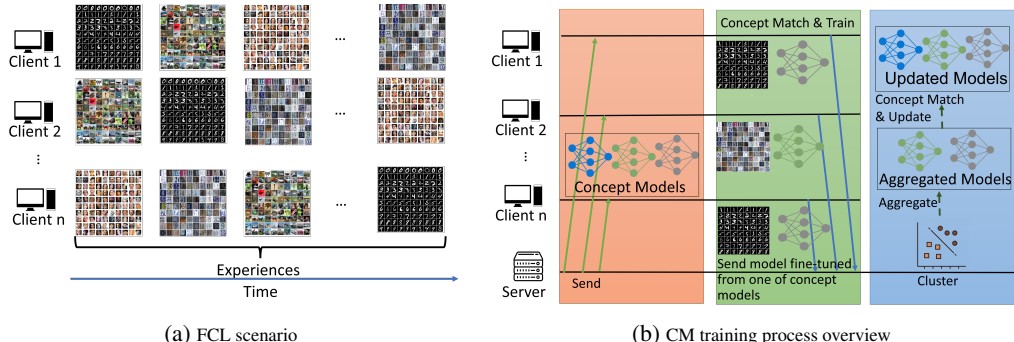

(a) FCL scenario            (b) CM training process overview

Figure 1: CM training for an FCL scenario

FCL under realistic assumptions. The main idea of CM is to separate the client models representing the same concept, and the server and the clients collaboratively train different concept models for each concept by matching the concept concealed in the data and the model. In the CM framework, the clients do not send any addition information beyond the model weights to the server, and thus the framework works for both class-incremental and task-incremental scenarios.

Figure 1b illustrates the CM training in one FCL round. In the bootstrapping phase, the system administrator at the server-side determines the input and output of the model, the estimated number of concepts in the data, and designs the neural network. In the initialization phase (**orange box**), the server transmits the global concept models to all clients. To avoid catastrophic forgetting, in the client operation phase (**green box**), each client utilizes its data in the current round to perform concept matching and chooses a best-matching model for further fine-tuning. Subsequently, the updated models are sent back to the server. To avoid interference among client models with different concepts, in the server operation phase (**blue box**), the server clusters the models with the same concept and aggregates them. Due to the fact that the union of the clients data in each round may not encompass all concepts, the number of clusters may not correspond to the number of concepts. The server does not know the concept captured by an aggregated cluster model. Thus, the server must perform concept matching to associate the aggregated cluster models with the global concept models of the previous round, and only update the relevant concept models for the next round.

The number of concepts is typically a small constant estimated from the semantics of the application, and the system administrator can select its value using domain knowledge. For example, a HAR model can use the locations of the user activities as different concepts, such as home, workplace, park, etc. Image data, on the other hand, can use the number of categories as the number of concepts. If an unexpected new concept occurs, by design, the system will automatically treat the new concept as the most similar existing concept.

The CM framework uses different concept matching algorithms at the server and the clients. In the server concept matching algorithm, for each global concept model, the server maintains a distance record between the current model and the model of the previous round. The server then matches the cluster aggregated model with a global concept model such that their distance is smaller than the record. This is to ensure the global concept model will be updated in the correct gradient descent direction. If there are multiple qualifying global concept models, the server selects the one closest in distance to the cluster model. It then updates the global concept model with the matching cluster model. For the client concept matching, an efficient algorithm tests the local data with the concept models and chooses the model with the lowest loss.

## 3   RELATED WORK

Most of the works on generic FL focus on system design (Jiang et al., 2022; McMahan et al., 2017; Beutel et al., 2020), model performance (Zhang et al., 2022; Gao et al., 2022; Jiang et al., 2023; Huang et al., 2022), privacy (Geiping et al., 2020; Wei et al., 2020), and communication and computation overhead (Jiang and Borcea, 2023; Wu et al., 2020; Luping et al., 2019). This paper does not focus on these generic FL challenges. Relevant for this paper are the works (Ouyang et al., 2022;

Ghosh et al., 2020; Sattler et al., 2020; Zhang et al., 2021) on clustering the client models in FL. These clustering approaches in FL assume the number of client groups is a constant, and cannot be applied directly in CL scenarios. All the works mentioned here assume the training data for the clients do not change over time, which limits the applicability of FL. Our work, on the other hand, focuses on making FL work well in the presence of dynamic changes of the concepts in data.

Continual Learning (CL) allows an AI model to learn continuously over time from a stream of data, while avoiding catastrophic forgetting. Recent works addressing CL can be categorized into three families (De Lange et al., 2021): replay (Rolnick et al., 2019; Isele and Cosgun, 2018), regularization (Ahn et al., 2019; Aljundi et al., 2019) and parameter isolation (Mallya and Lazebnik, 2018; Serra et al., 2018). These techniques do not address additional challenges from FL. In addition to its distributed nature, FL also introduces privacy restrictions. For example, FL clients shall not share their task IDs with the server in the task-incremental CL scenario. In addition, even if the clients can learn new concepts well without forgetting the previous ones, the aggregation may sabotage the efforts of the clients when their learning paths diverge due to non-iid data. This phenomenon has been demonstrated experimentally with image data in a recent work (Yoon et al., 2021). Expanding CL to FL, our work adheres to the FL requirement that the server only accesses the client model weights, and it handles the interference among the clients in FL.

Federated Continual Learning (FCL) is a newly introduced research area that combines FL and CL. FedWeIT (Yoon et al., 2021) and CFeD (Ma et al., 2022) only work in the task-incremental scenario, and they requires the server to know the task IDs from the clients. CDA-FedAvg (Casado et al., 2022) only considers the context information (i.e. the positions of sensors in HAR) as the concepts, and it is not proven to work with concept drift caused by different sets of classes. Other works (Guo et al., 2021; Dong et al., 2022; Qi et al., 2023; Zhang et al., 2023) do not address the potential interference among the clients. Unlike these works, our work tackles catastrophic forgetting and the interference among the clients under more realistic assumptions, such as the clients do not share any additional information with the server beyond the model weights, and the classes can change over time.

## 4 CM FRAMEWORK

This section describes the problem definition, the CM learning framework, several design choices, and the concept matching algorithms.

### 4.1 PROBLEM DEFINITION

For each client $n \in \{1, 2, ..., N\}$, the data arrives in a streaming fashion as a (possible infinite) sequence of learning experiences $\mathcal{S}_n = e_n^1, e_n^2, ..., e_n^t$. Without loss of generality, each experience $e_n^t$ consists a batch of samples $\mathcal{D}_n^t$, where the $i$-th sample is a tuple $\langle x_i, y_i \rangle_n^t$ of input and target respectively. Let $\mathcal{C} = \{C_1, C_2, ..., C_k\}$ be the set of $K$ concepts hidden in entire dataset $\mathcal{D}$. Each concept $C_k$ is associated with a probability distribution $P_k(X, Y)$, where $X$ denotes the input space and $Y$ denotes the label space. A batch of client samples follows one of the distributions $\mathcal{D}_n^t \sim P_k(X, Y)$, which may or may not be explicitly known by the client.

The goal of the CM framework is to learn a set of models $\{w_k\}_{k=1}^K$, and each model $w_k$ can perform well for its corresponding concept $C_k$. The problem can be formulated as the Eq. 1, where $L$ is the loss function, $\mathcal{D}_n$ is the entire stream of data on client $n$.

$$\arg \min_{\{w_k\}_{k=1}^K} \sum_{k=1}^K \sum_{n=1}^N L(w_k, \mathcal{D}_n) \tag{1}$$

### 4.2 LEARNING FRAMEWORK FOR FCL

To begin the learning process, the system administrator at the server side determines the number of concepts $K$ and designs the model. The server initializes the weights of $K$ global concept models randomly and sends them to the clients. In every round, each client $n$ receives the weights of global concept models $W^{t-1} = (w_1, w_2, ..., w_K)^{t-1}$ from the server. To avoid catastrophic forgetting

caused by training a model with the data of different concepts, the clients perform concept matching with the local data of current round to select the best-matching global concept model as Eq. 2.

$$k_n^* = ClientConceptMatch(W^{t-1}, \mathcal{D}_n^t) \tag{2}$$

Next, the client fine-tunes the best-matching global concept model weights $w_{k^*}$ with the local data, and produces a new local model with weights $\theta_n^t$ and sends it the server. After receiving the client models with weights $\{\theta_n^t\}_{n=1}^N$, to separate the clients models with the different concepts, the server clusters the client models into a set of clusters of size $J$, denoted as $\Omega^t$ (Eq. 3). Since the union of clients data per round may not cover all concepts, $J$ and $K$ are usually different.

$$\Omega^t = Cluster(\{\theta_n^t\}_{n=1}^N) \tag{3}$$

$$W'^t = Aggregate(\Omega^t) \tag{4}$$

Then, the server produces aggregated cluster models with weights $W'^t = (w'_1, w'_2, ..., w'_J)^t$ (Eq. 4). The server does not know the concept in a aggregated cluster model or which global concept model to update. Therefore, it need match the aggregated cluster models $\Theta^t$ with the global concept models $W^{t-1}$, and only update the global concept models with data encountered in this round as Eq. 5.

$$W^t = ServerConceptMatch(W'^t, W^{t-1}) \tag{5}$$

The pseudo-code of the CM framework is described in Appendix A.

## 4.3 Design Discussion

The CM framework provides the flexibility to use different clustering, aggregation, and concept matching algorithms. It can evolve, as new algorithms are proposed for different applications and models. The aggregation algorithms in FL are orthogonal to the CM framework, and any of them can be employed to further mitigate non-iid. We evaluate the CM framework with multiple clustering algorithms in section 5.2, and demonstrates that CM works well with classic clustering algorithms, such as kmean, agglomerative, and BIRCH. Some clustering algorithms, such as DBSCAN (Ester et al., 1996) and OPTICS (Ankerst et al., 1999), do not require the number of clusters to be known. As a future work, the CM framework may be extended to work in such a scenario. In addition, dimension reduction techniques (Carreira-Perpinán, 1997) can also be applied in conjunction with clustering algorithms to mitigate the curse of dimensionality.

The CM framework is designed to be compatible with both task-incremental and class-incremental scenarios, because the clients do not need to possess any prior understanding of concept drift in data, including task IDs. In both scenarios, the server does not know the concepts of clients data, and it shall perform clustering and concept matching every round. In the task-incremental scenario where the clients know the task IDs, the clients only need to perform concept matching at the initial rounds. The clients can maintain a mapping between the global concept model ID and the task ID. After the client encounters all tasks, it can just use the mapping to match the data of a task with the global concept model. During inference, the clients can pick the corresponding global concept model to perform prediction with their input data. In the class-incremental scenario, the clients have to perform concept matching every round. At inference, the clients can apply an ensemble model method (Dietterich, 2000) to produce an output from all the global concept models.

Regarding privacy, the CM framework is the same as vanilla FL (i.e., clients send only their model weights to server). An adversarial server might seek to extract information from the client model using the same techniques from vanilla FL, which is beyond the scope of this paper. Regarding communication, the clients send a single model trained with the local data in the same way as vanilla FL, but the server sends multiple concept models to the clients. The number of concepts is usually a small constant under the control of the system administrator, and the concept models designed in CM can be smaller than the single model in vanilla FL, because each model only learns a single concept. Nevertheless, it is essential to balance the trade-off between model performance and communication

overhead, by taking into account the available system resources. To further improve privacy protection and communication efficiency, CM can use existing privacy protection (Mothukuri et al., 2021) or communication reduction (Deng et al., 2020) techniques for FL.

## 4.4 CONCEPT MATCHING ALGORITHMS

The concept matching algorithms at the client and the server collaboratively and iteratively update each global concept model with the information learnt from the data of a matching concept, and achieve up to 100% accuracy. Two algorithms are connected through client training, and server clustering and aggregation. The main novelty of this distributed approach lies in the server concept matching algorithm, which ensures the model updates in the correct gradient descent direction.

---

**Algorithm 1** Server Concept Matching Pseudo-code

---

1: **procedure** SERVERCONCEPTMATCH($W', W$)
2:     // Executed at Server
3:     require $distRecord$ of size $K$ as the global record of distance between each global concept model and the corresponding previous global concept model
4:     **for** each aggregated cluster model $w'_j \in W'$ **do**
5:         $candidate \leftarrow null$
6:         $candidateDist \leftarrow \infty$
7:         **for** each global concept model $w_k \in W$ **do**
8:             $tmpDist \leftarrow Distance(w'_j, w_k)$
9:             **if** $tmpDist < distRecord[k]$ and $tmpDist < candidateDist$ **then**
10:                 $candidate \leftarrow k$
11:                 $candidateDist \leftarrow tmpDist$
12:         $W[candidate] \leftarrow w'_j$
13:         $distRecord[candidate] \leftarrow candidateDist$
14:     **return** $W$

---

**Server Concept Matching.** After clustering, the groups of the client models fine-tuned with the data of different concepts are unordered, and the number of clusters may not be the same as the total number of concepts because the union of the clients' data in the current round may not cover all concepts. The server does not know how to update the global concept models without the matching between the aggregated cluster models and the global concept models from previous round.

To resolve this challenge, we propose a novel distance-based server concept matching algorithm. This algorithm not only updates a global concept model with a close cluster model in distance, but also ensures the update in the correct gradient descent direction. Our algorithm can use different distance metrics. For a normal size neural network such as LeNet (LeCun et al., 1998), Manhattan or Euclidean distance can be employed for their low computational complexity. For larger neural networks, the dimension reduction techniques (Carreira-Perpinán, 1997) can be incorporated to mitigate the curse of dimensionality.

The pseudo-code of server concept matching algorithm is shown in Algorithm 1. The algorithm requires a global record of the distance between each global concept model and the corresponding previous global concept model (line 3). For each aggregated cluster model (line 4), the algorithm tracks its best-matching candidate (line 5) and its distance from the best-matching candidate (line 6). Each aggregated cluster model is compared with each global concept model (line 7) by computing their distance (line 8). If their distance is smaller than both the global distance record of the corresponding concept $k$ and the distance from previous matching candidate (line 9), we consider them a better match (line 10) and update the distance between the cluster model and its matching candidate (line 11). After checking all the global concept models, the algorithm updates the best-matching global concept model with the aggregated cluster model (line 12), and also updates the global distance record with the distance between the best-matching pair (line 13).

This algorithm is theoretically grounded. Intuitively, in a gradient descent-based learning algorithm, as the learning curve becomes flatter over iterations, the learning progress slows down. Therefore, the distance between the current model and the model of the previous iteration becomes smaller. Theorem 1 formulates this intuition, and Algorithm 1 utilizes this theorem. By tracking and com-

paring with the distance record, Algorithm 1 updates each concept model with a matching cluster model only when their distance becomes smaller than the current distance. The proof of Theorem 1 demonstrates theoretically that Algorithm 1 updates the global concept model with a matching cluster model in the correct gradient descent path. This allows the concept models to learn over time without interference from other concepts.

**Assumption 1.** *Differentiability: The loss function $L(w)$, used to optimize a neural network, is differentiable with respect to the model parameters $w$.*

**Assumption 2.** *Lipschitz continuity: The gradient of the loss function $\nabla L(w)$ is Lipschitz continuous with a positive constant L. By Lipschitz continuity definition, for any two points $w^1$ and $w^2$, the following inequality holds $\|\nabla L(w^1) - \nabla L(w^2)\| \leq L\|w^1 - w^2\|$, where $\|.\|$ denotes the norm.*

**Theorem 1.** *Given a loss function $L(w)$ under assumptions 1 and 2, $w$ is updated with gradient descent $w^{t+1} = w^t - \eta \nabla L(w^t)$, where $t$ is the iteration number, $\eta$ is the learning rate, and $\nabla L(w^t)$ is the gradient of the loss function with respect to $w^t$, the following inequality holds $\|w^{t+1} - w^t\| < \|w^t - w^{t-1}\|$.*

Theorem 1 is under Assumption 1 Differentiability and Assumption 2 Lipschitz continuity. The proof of the theorem is in Appendix B. Without requiring strong assumptions, such as convexity, Assumption 1 and 2 can be applied to most loss functions for neural networks. In the theorem, gradient descent is also the prevalent algorithm to update neural networks. Therefore, this theorem can be applied to the general optimization process of most neural networks.

**Client Concept Matching.** The clients receive the global concept models from the server every round, and each model shall learn the data distribution of each concept. The clients need to select one of the global concept models, and fine-tune it with the data of the current round. Since the clients may or may not be aware of the concepts, they shall perform client concept matching to match the concept of the current data with a global concept model.

At round $t$, a client $n$ can test the global concept models of the previous round $\{w_k^{t-1}\}_1^K$ on its current local data $\mathcal{D}_n^t$, and select the $k^*$-th concept model with the smallest loss for further fine-tuning as Eq. 6. Since the data do not accumulate over a given limit in FCL, testing the models is an effective method to select the best-matching global concept model without significant overhead.

$$k^* = \arg\min_k L(w_k^{t-1}, \mathcal{D}_n^t) \tag{6}$$

## 5 EVALUATION

The evaluation has five main goals: (i) Investigate CM effectiveness; (ii) Compare with the baselines; (iii) Quantify the performance of different clustering algorithms; (iv) Quantify the concept matching accuracy; (v) Investigate the scalabilitiy in terms of the number of clients and the model size. In the Appendix C.6, we evaluate CM with diverse experimental settings, demonstrate CM has good resilience when configured with numbers of concepts that are different from the ground truth. We also evaluate CM on a Qualcomm QCS605 IoT device, and demonstrate its feasibility in real-world and its low overhead in terms of the client end-to-end operation time.

### 5.1 EXPERIMENTAL SETUP

Similar to a recent FCL work (Yoon et al., 2021), we evaluate CM with a "super" dataset, which consists of six frequently used image datasets: SVHN, FaceScrub, MNIST, Fashion-MNIST, Not-MNIST, and TrafficSigns. To simulate different concepts, the "super" dataset is split into five concepts. More dataset details are described in Appendix C.1.

Non-overlapping chunks from the five concept datasets are further distributed randomly to the clients. At every round, the local data across clients are non-IID. Each client encounters one of the five local concept datasets randomly, and uses a cyclic sliding window of 320 samples in the encountered concept dataset. Unless otherwise specified, CM is tested with 20 clients (all clients participate in each training round), kmean clustering algorithm, FedAvg aggregation algorithm, and Manhattan distance for the server concept matching algorithm. To compare with the baseline fairly, we use the same CNN-based model as (Yoon et al., 2021), detailed in Appendix C.2. We test CM

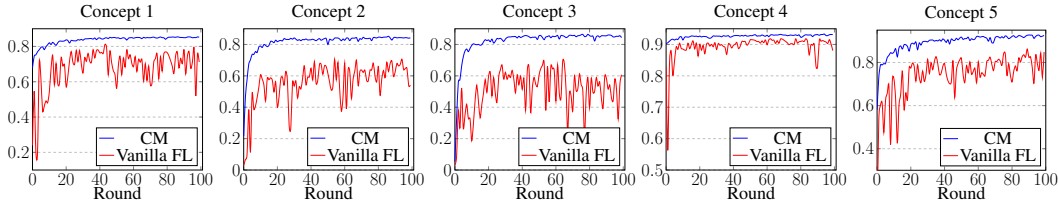

Figure 2: CM vs. vanilla FL: Test set accuracy over communication rounds

Table 1: Test set accuracy (%) comparison with baselines

| Concept | 1 | 2 | 3 | 4 | 5 | avg |
|---|---|---|---|---|---|---|
| EWC | 82.3 | 74.5 | 72.3 | 92.1 | 85.9 | 86.7 |
| FedWeIT | 61.0 | 62.0 | 66.8 | 72.6 | 70.7 | 68.3 |
| CM | 85.4 | 85.2 | 86.5 | 93.3 | 92.4 | 90.3 |

with different hyper-parameters, and only present the results with the hyper-parameters that lead to the best results. The system runs 100 rounds of training for each experiment. More details of Python libraries used, computing equipment and hyper-parameters are described in Appendix C.3.

To quantify the overall concept matching accuracy, we evaluate under the class-incremental scenario, and assume the clients are not aware of the concepts and perform the client concept matching every round. In task-incremental scenarios, the difference is that the clients do not have to perform concept matching every round, because they know the task IDs and the concept drift due to the transition of different tasks. For the same experimental setting, task-incremental training would achieve the same model performance (Appendix C.6 Figure 4) with lower computation overhead at the clients.

## 5.2 RESULTS

**Effectiveness of CM in FCL vs. Vanilla FL.** Figure 2 demonstrates the effectiveness of CM, as the volatile learning curves of vanilla FL prominently illustrate the catastrophic forgetting due to the concept drift and the potential interference among clients. CM can effectively eliminate these negative effects with smooth learning curves for all concepts. The superior performance of CM is more apparent for difficult concepts (RBG images with 50 classes from FaceScrub in Concepts 2 and 3) than in easy concepts (BW images with 30 classes from mixed MNIST in Concept 4). Compared with vanilla FL, the concept model accuracy improvement is up to 17.5%, and the weighted average over the number of samples for all the concepts is also improved from 86.2% to 90.3%. Considering the complexity of the super dataset, this improvement is significant for image classification.

**Comparison with FCL baselines.** Table 1 shows the results of the comparison, which is performed with the same experimental setup. EWC (French, 1999) is a commonly employed baseline in CL. We apply it on the clients' training, and then the server aggregates with FedAvg. FedWeIT (Yoon et al., 2021) is a recently published work for FCL. CM outperforms the baselines and achieves 90.3% accuracy. While EWC performs reasonably well (86.7%), FedWeIT does not perform well under more realistic assumptions and in a larger scale experiment than its original evaluation (i.e., 5 clients per round, 5 classes to train per client, and non-overlapping classes over clients). Since FedWeIT applies a completely different design when learning information across tasks or concepts, its inferior performance may be partially due to the sparse parameters employed, which fail to separate different concepts and fully capture the complex information (i.e. up to 50 classes) in the concepts.

**Performance of clustering algorithms.** Table 2 shows the results for 5 clustering algorithms whose parameters are detailed in Appendix C.5. A perfect clustering can group all client models correctly with the same concept. Adjusted Rand Index (ARI) is a commonly used metric for clustering algorithms: 1.0 stands for perfect matching. Table 2 also shows the minimum ARI, as the worst clustering performance over 100 rounds. The results show that CM with BIRCH performs best, as it achieves up to 96 rounds of perfect clustering out of 100 and 0.994 average ARI. Furthermore, all algorithms perform reasonably well and achieve over 88.4% average model accuracy.

Table 2: Clustering performance with 100 rounds training

|  | Kmean | Agglomerative | BIRCH | DBSCAN | OPTICS |
|---|---|---|---|---|---|
| Rounds with perfect clustering | 91 | 93 | 96 | 66 | 50 |
| ARI (average) | 0.988 | 0.989 | 0.994 | 0.972 | 0.888 |
| ARI (minimum) | 0.713 | 0.704 | 0.771 | 0.700 | 0.478 |
| Model accuracy % (average) | 90.3 | 90.1 | 90.4 | 88.5 | 88.4 |

Table 3: Concept matching accuracy (%) with 100 training rounds

|  | Kmean | Agglomerative | BIRCH | DBSCAN | OPTICS |
|---|---|---|---|---|---|
| Manhattan | 100 | 100 | 100 | 97.4 | 93.4 |
| Euclidean | 100 | 99.8 | 100 | 90.4 | 94.9 |
| Chebyshev | 98.6 | 99.9 | 100 | 97.7 | 93.6 |

Table 4: CM performance vs. number of clients

|  | ARI (average) | CM accuracy % | Model accuracy % |
|---|---|---|---|
| 20 | 0.988 | 100 | 90.3 |
| 40 | 0.983 | 99.7 | 95.3 |
| 80 | 1.000 | 100 | 95.4 |

Table 5: CM performance vs. model size

|  | ARI (average) | CM accuracy % | Model accuracy % |
|---|---|---|---|
| -20% | 0.983 | 100 | 90.0 |
| original | 0.988 | 100 | 90.3 |
| +20% | 0.999 | 100 | 90.4 |

**Concept matching accuracy.** The concept matching accuracy is defined as the percentage of correct concept matching over the entire training process (i.e., the collaborative client/server matching). Table 3 shows the concept matching accuracy with a variety of clustering algorithms and distance metrics. To quantify the concept matching accuracy separately with minimum influence from the clustering errors, we use the cluster ID that has the most number of clients with the given data concept, instead of the actual cluster ID for the mapping of a given client data concept. The results demonstrate that CM achieves up to 100% concept matching accuracy. Table 2 and 3 demonstrate that CM provides the flexibility to use different clustering algorithms and distance metrics, as it performs well with all of them. As neural networks grow in size and complexity, the curse of dimensionality may manifest itself and requires advanced clustering algorithms and distance metrics.

**Concept matching scalability.** Table 4 shows CM performs well as the number of clients increases. With 80 clients, both the clustering and the CM algorithms perform perfectly. This is because the clustering algorithms generally perform better with larger number of samples. CM enjoys this benefit and may achieve better model performance (up to 95.4%) with a larger number of clients. We further test CM with 20% increase or decrease in the size of CNN layer channels and dense layer neurons. A larger model can further stress-test CM, and a smaller model can reduce the communication overhead for CM. To avoid overfitting the model under the given dataset, we could not further downsize the model or split the data into more clients. Table 5 shows the performance metrics of CM, and there is a small improvement (90.4%) in accuracy when using a larger model. Tables 4 and 5 show that both the clustering and the concept matching achieve near flawless performance.

## 6 CONCLUSION

Concept Matching (CM) is a novel FCL framework to alleviate catastrophic forgetting and interference among clients by training different models for different concepts concealed in the data. To avoid interference among clients, CM uses a clustering algorithm to group the client models with the same concept. To mitigate catastrophic forgetting, the server and the clients run concept matching algorithms that collaboratively train and update each concept model with the matching data of the same concept. Also, the server concept matching algorithm ensures the updating of the concept model in the correct gradient descent direction. CM achieves higher model accuracy than state-of-the-art systems, and works regardless of whether the clients are aware of the concepts or not. Our extensive evaluation also demonstrates that CM performs well with a variety of clustering algorithms and distance metrics, and scales well with the number of clients and the model size.

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

## A   ALGORITHMIC DESCRIPTION OF CONCEPT MATCHING FRAMEWORK

Algorithm 2 describes the pseudo-code of the CM framework. CM executes as a multi-round, iterative FL cycle (lines 3-9). At each round (line 3), each client operates in parallel (line 5), including clients concept matching (line 13), local model updates with batches of data (line 14-16), and returning the client model weights to the server (line 18). Then, server performs clustering (line 7), aggregation (line 8), and server concept matching to update the global concept models with the aggregated cluster models (line 9).

---

**Algorithm 2** Concept Matching Framework Pseudo-code

---

1: **procedure** SERVEREXECUTE:
2:     initialize $W^0 = (w_1, w_2, ..., w_K)^0$ **randomly**, total number of rounds $T$
3:     **for** $t = 1$ to $T$ **do**
4:         // Update Done at Clients and Returned to Server
5:         **for** each client $n$ **do** // In Parallel
6:             $\theta_n$ = n.CLIENTUPDATE($W^{t-1}$)
7:         $\Omega^t = Cluster(\{\theta_n^t\}_{n=1}^N)$
8:         $\Theta^t = Aggregate(\Omega^t)$
9:         $W^t = ServerConceptMatch(\Theta^t, W^{t-1})$

10: **procedure** CLIENTUPDATE($W$)
11:     // Executed at Clients
12:     require step size hyperparameter $\eta$, local dataset of current round $\mathcal{D}$
13:     $k^* = ClientConceptMatch(W, \mathcal{D})$
14:     $x_n \leftarrow \mathcal{D}$ divided into minibatches
15:     **for** each batch $b \in x_n$ **do**
16:         $\theta_n = w_{k^*} - \eta \nabla L(w_{k^*}, b)$
17:     // Results Returned to Server
18:     **return** $\theta_n$

---

## B   THEOREM PROOF

**Assumption 1.** *Differentiability: The loss function $L(w)$, used to optimize a neural network, is differentiable with respect to the model parameters $w$.*

**Assumption 2.** *Lipschitz continuity: The gradient of the loss function $\nabla L(w)$ is Lipschitz continuous with a positive constant L. By Lipschitz continuity definition, for any two points $w^1$ and $w^2$, the following inequality holds $\|\nabla L(w^1) - \nabla L(w^2)\| \leq L\|w^1 - w^2\|$, where $\|.\|$ denotes the norm.*

**Lemma 1.** *Given a loss function $L(w)$ under assumption 1 and 2, $w$ is updated with gradient descent $w^{t+1} = w^t - \eta \nabla L(w^t)$, where $t$ is the iteration number, $\eta$ is the learning rate, and $\nabla L(w^t)$ is the gradient of the loss function with respect to $w^t$, the following inequality holds $\|\nabla L(w^{t+1})\| < \|\nabla L(w^t)\|$.*

*Proof.* To prove $\|\nabla L(w^{t+1})\| < \|\nabla L(w^t)\|$, we can use the assumption 2. Let L be the Lipschitz constant. Then, we have $\|\nabla L(w^{t+1}) - \nabla L(w^t)\| \leq L\|w^{t+1} - w^t\|$.

Now, using the reverse triangle inequality, we can write $\|\nabla L(w^{t+1})\| - \|\nabla L(w^t)\| \leq \|\nabla L(w^{t+1}) - \nabla L(w^t)\|$. Substituting the previous inequality, we get $\|\nabla L(w^{t+1})\| - \|\nabla L(w^t)\| \leq L\|w^{t+1} - w^t\|$.

Using the gradient descent update $w^{t+1} = w^t - \eta \nabla L(w^t)$, we can write $\|\nabla L(w^{t+1})\| - \|\nabla L(w^t)\| \leq L\eta\|\nabla L(w^t)\|$. Rearranging the terms, we get $\|\nabla L(w^{t+1})\| \leq (1 - L\eta)\|\nabla L(w^t)\|$.

Since L and $\eta$ are both positive, we have $1 - L\eta < 1$. Therefore, we can conclude that $\|\nabla L(w^{t+1})\| < \|\nabla L(w^t)\|$. This completes the proof. □

**Theorem 1.** *Given a loss function $L(w)$ under assumptions 1 and 2, $w$ is updated with gradient descent $w^{t+1} = w^t - \eta \nabla L(w^t)$, where $t$ is the iteration number, $\eta$ is the learning rate, and $\nabla L(w^t)$ is the gradient of the loss function with respect to $w^t$, the following inequality holds $\|w^{t+1} - w^t\| < \|w^t - w^{t-1}\|$.*

*Proof.* From the inequality in lemma 1, $\|\nabla L(w^t)\| < \|\nabla L(w^{t-1})\|$, using the gradient descent update $w^{t+1} = w^t - \eta\nabla L(w^t)$, we write $\|w^{t+1} - w^t\| < \|w^t - w^{t-1}\|$. This completes the proof. $\square$

## C  EVALUATION DETAILS

### C.1  DATASET

We evaluate CM with a "super" dataset, similar to a recent FCL work (Yoon et al., 2021), which consists of six frequently used image datasets: SVHN, FaceScrub, MNIST, Fashion-MNIST, Not-MNIST, and TrafficSigns. To simulate different concepts, the "super" dataset is splitted into five concepts. As shown in Table 6, the data in the five concepts differ across a wide spectrum of classes and number of samples. The original FaceScrub dataset has 100 classes. In order to stress test CM, it is splitted into Concept 2 and 3 with 50 different classes each, as we aim to verify whether CM can differentiate them successfully. Since MNIST datasets are easy to learn, we mix the three MNIST datasets together to make it more difficult. The training, test, and validation split of the data follows 7:2:1.

Table 6: Dataset details for each concept

| Concept | 1 | 2 | 3 | 4 | 5 |
|---|---|---|---|---|---|
| Dataset | SVHN | FaceScrub0 | FaceScrub1 | MNIST, Fashion-MNIST, Not-MNIST | TrafficSigns |
| No. Classes | 10 | 50 | 50 | 30 | 43 |
| No. Samples | 88300 | 9898 | 9899 | 138197 | 45956 |

### C.2  MODEL

To compare with the baseline fairly, we use the same CNN-based image classification model as (Yoon et al., 2021). We believe this model is ideal in size to learn from the dataset. This model uses two convolutional layers and three dense layers to classify an image input (32*32*3) into one of the 183 classes. The two convolutional layers have 20 and 50 channels, with 5 by 5 filters, stride of 1, and ReLU activation. A 3 by 3 max pooling with stride of 2 follows them. Then, the flattened tensor is fed into three dense layers of 800, 500 and 183 neurons respectively, with ReLU and Softmax activation.

As it is common practice in class-incremental CL, the model follows the single-head evaluation setup (Shim et al., 2021; Mai et al., 2021; 2022), where it has one output head to classify all labels. This setup is ideal for clients with constrained resource capacity in FL, because the clients do not have to spend computation resources on expanding or selecting the output head. Let us note that using the total number of labels as the model output size does not mean we have to know the entire label space or even the label space size, because we can use any output size not smaller than the upper bound of the number of labels. It makes no difference in the training and testing accuracy when experimenting with a given dataset, because the weights associated with unencountered output neurons will not be updated in the backward pass.

### C.3  EXPERIMENTAL SETTINGS

We implement CM with TensorFlow and scikit-learn. The experiments are conducted on a Ubuntu Linux cluster (Intel(R) Xeon(R) CPU E5-2680 v4 @ 2.40GHz with 512GB memory, 2 NVIDIA P100-SXM2 GPUs with 16GB total memory). Non-overlapping chunks from the five concept datasets are further distributed randomly to the clients. At every round, the local data across clients are non-IID. Each client encounters one of the five local concept datasets randomly, and uses a cyclic sliding window of 320 samples in the encountered concept dataset. Unless otherwise specified, CM is tested with 20 clients (all clients participate in each training round), kmean clustering algorithm, FedAvg aggregation algorithm, and Manhattan distance for the server CM algorithm. For the local training, we use the Adam optimizer with learning rate of 0.001, weight initializer of HeUniform, and batch size of 64. Each client maintains and uses a single Adam optimizer throughout the training for all concepts. We train 15 epochs every round, and use early stopping with the patience value as 3.

We test CM with different hyper-parameters, and only present the results with the hyper-parameters that lead to the best results. The system runs 100 rounds of training for each experiment.

To quantify the overall concept matching accuracy, we evaluate under the class-incremental scenario, and assume the clients are not aware of the concepts and perform the client concept matching every round. In task-incremental scenarios, the difference is that the clients do not have to perform concept matching every round, because they know the task IDs and the concept drift due to the transition of different tasks. For the same experimental setting, task-incremental training would achieve the same model performance with lower computation overhead at the clients.

## C.4    IoT Device Setup

CM is evaluated on a Qualcomm QCS605 IoT device. This device is equipped with Snapdragon™ QCS605 64-bit ARM v8-compliant octa-core CPU up to 2.5 GHz, Adreno 615 GPU, 8G RAM, and 16 GB eMMC 5.1 onboard storage. We choose it because its specifications are ideal for AIoT cameras and image-based applications. The device is connected to Internet through WiFi with a bandwidth of 300 Mbps. We re-use CM simulation Python code on the device. Since the device does not support native Linux and its the operating system is rooted Android 8.1, we need to run a Linux distribution on the Linux kernel of Android for easy package management and better support. We achieve this goal with two open source projects: termux-app and ubuntu-in-termux. Termux-app is an Android application for terminal application and Linux environment. It provides some basic Linux commands and packages, but is not on par with a mature Linux distribution, such as Ubuntu. Ubuntu-in-termux bridges the gap. Through it, we are able to install well-maintained Python environments and libraries to execute training on-device. The Python training process can be observed directly under adb shell, which does not include the overhead of Termux-app or Ubuntu-in-termux.

## C.5    Parameters for Clustering Algorithms

Kmean, agglomerative, and BIRCH require the number of clusters as a parameter. Unless otherwise specified, we set this parameter to be 5. DBSCAN and OPTICS require some threshold values tuned for the data as parameters instead of the number of clusters. We adhere to the convention when selecting their parameters. For DBSCAN, there are two main parameters: $min\_samples$ and $\epsilon$. $min\_samples$ is the fewest number of points required to form a cluster. We adjust it to be 3, which is a lower value than the average number of clients per concept (20/5). $\epsilon$ is the maximum distance between two points while the two points can still belong to the same cluster. To choose $\epsilon$, we firstly calculate the average distance between each point (the model weights of a client) and its 3 ($min\_samples$) nearest neighbors, and then we sort distance values in the ascending order and plot them. We choose $\epsilon$ to be 20 as the point of maximum curvature in the plot. Similarly, we set $min\_samples$ parameter to be 3 for OPTICS. The other parameters for these clustering algorithms are the default values in scikit-learn.

## C.6    Additional Results

**CM vs. Vanilla FL with each dataset as a concept.** We further diversify the experimental settings by treating each dataset as a concept without mixing or splitting the datasets. To evaluate CM comprehensively, we compare the test set accuracy between vanilla FL and CM. Similar to the original experimental settings, Figure 3 illustrates the smooth learning progress for CM over vanilla FL for all concepts. Compared with vanilla FL, the weighted average accuracy over the number of samples for all the concepts is improved from 85.5% to 88.0% as well.

**Learning curves for scalability.** Figure 5 shows the learning curves as the number of clients increases. The results show that CM can always learn smoothly. Figure 6 demonstrates CM can also learn smoothly with 20 clients as the model size increases or decreases 20%. To stress test CM, we further investigate the model performance when training 80 clients with the network size increased and decreased 20%. The learning curves in Figure 7 demonstrate CM's smooth learning progress, as it achieves 94.3% and 94.9% average accuracy, respectively.

**Resilience to number of concepts different from the ground truth.** CM requires an estimated number of concepts configured in the bootstrapping phrase. In case the system administrator fails

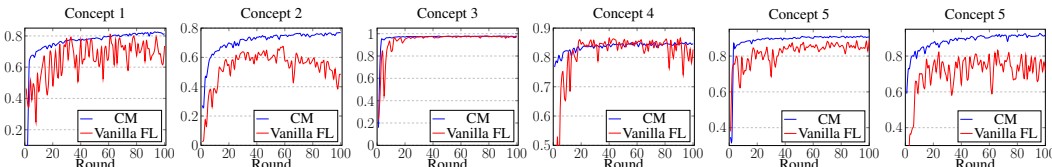

Figure 3: CM vs. vanilla FL: Test set accuracy over communication rounds

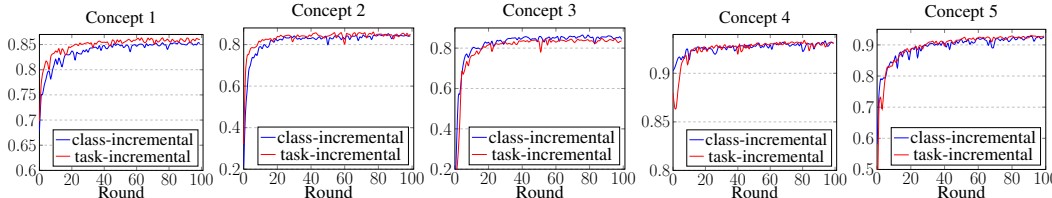

Figure 4: Class-incremental vs. task-incremental: Test set accuracy over communication rounds

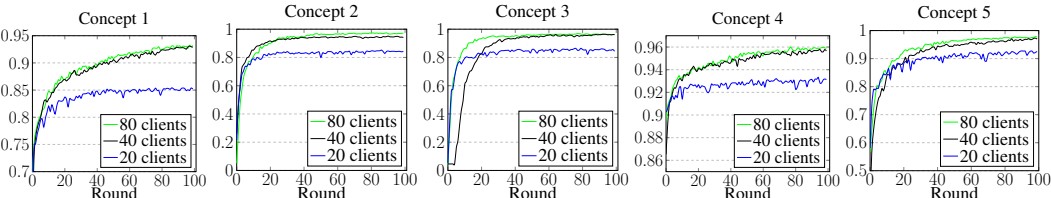

Figure 5: Test set accuracy vs. communication rounds as number of clients increases

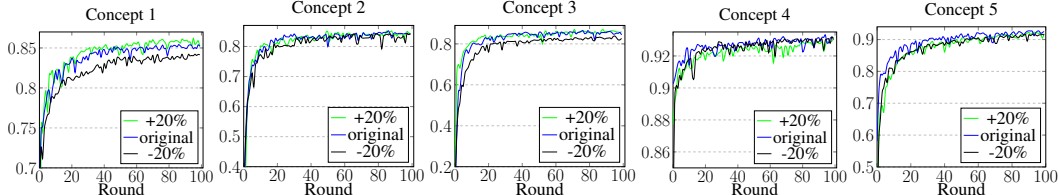

Figure 6: Test set accuracy vs. communication rounds for training 20 clients with different model size

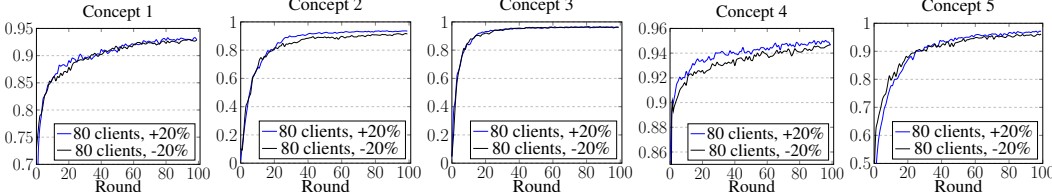

Figure 7: Test set accuracy vs. communication rounds for training 80 clients with different model size

to estimate the number of concept correctly, we experiment with the number of concepts from 3 to 7 under the same experimental settings (i.e., 5 concepts is the ground truth). As shown in Figure 8, when the configured number of concepts is higher (6 and 7) than the ground truth, 5 concept models (out of 6 or 7) learn the corresponding concepts smoothly. The extra concept models in the system do not effect the smooth learning progress, and the average model accuracy achieves 90.5% and 90.0% respectively. When the number of concepts is smaller (4 and 3) than the ground truth, the system starts to treat the similar concepts (e.g., two FaceScrub concepts) as one. Although the model accuracy on the affected concepts (2, 3, and 5) exhibit minor fluctuations during the training, the

Figure 8: Test set accuracy with different number of concepts over communication rounds

Table 7: Client operation time (second) on real IoT device

|  | Receiving model | Concept Matching | Training | Sending model | Total |
|---|---|---|---|---|---|
| Vanilla FL | 0.67 | N/A | 85.27 | 0.67 | 86.61 |
| CM | 3.35 | 8.52 | 85.27 | 0.67 | 97.81 |

smooth learning progress for the other concepts (1 and 4) is not affected. Nevertheless, the average model accuracy achieves 89.5% and 88.9% respectively, and beats vanilla FL (86.7%). These results demonstrate CM has good resilience in terms of the number of concepts configured in the system. Furthermore, they suggest it is better if system administrators over-estimate the number of concepts, because the performance is still very good in this case.

**Client operation overhead.** Designed for mobile or IoT devices, CM is evaluated on a real IoT device in terms of the client end-to-end operation time. Compared with vanilla FL, the client operation overhead comes from receiving multiple concept models from the server and the client concept matching. Table 7 shows the breakdown of client operation time on the Qualcomm QCS605 IoT device in one round. We assume the worst-case scenario that the clients do not know the concepts, and perform concept matching every round. Compared with a multi-epoch training process over the entire experienced data, the concept matching can be achieved by testing only a portion of the data. The communication time is calculated as sending or receiving the model(s) of size 25.2MB over 300 Mbps WiFi network. The total client end-to-end operation time in one round is 97.81 seconds, which is feasible for a real-world deployment. Overall, the total CM operation time has a low overhead (11%) over vanilla FL operations on the IoT device. We believe the improvement in performance achieved by CM is worth this overhead cost.

