# OpenReview forum: "Concept Matching: Clustering-based Federated Continual Learning"
_ICLR.cc/2024/Conference — Submitted to ICLR 2024_

### Official Review · Reviewer_anyU · 2023-10-28

**Soundness:** 2 fair
**Presentation:** 2 fair
**Contribution:** 1 poor
**Rating:** 3
**Confidence:** 4

**Summary:**

This paper focuses on federated continual learning (FCL) and proposes Concept Matching (CM), a clustering-based FCL framework. CM learns different global models for different concepts and designs server/client matching to identify target concept models for local optimization and global aggregation.

**Strengths:**

S1. The idea of learning a model set for clients to select in FL setting is novel.

S2. The introduction of two main challenges in FCL is clear.

**Weaknesses:**

W1. There are some more related works [1, 2] about clustered federated learning not included in this paper.

W2. The authors cite several related works about FCL but compare only one of them in experiments.

W3. This paper claims that CM can handle both class-incremental and task-incremental FCL but the experiments only include class-incremental setting.

[1] Felix Sattler, Klaus-Robert Müller, and Wojciech Samek. 2021. Clustered Federated Learning: Model-Agnostic Distributed Multitask Optimization under Privacy Constraints. IEEE Trans. Neural Netw. Learn. Syst. 32, 8 (2021), 3710–37

[2] Yu Zhang, Moming Duan, Duo Liu, Li Li, Ao Ren, Xianzhang Chen, Yujuan Tan, and Chengliang Wang. 2021. CSAFL: A Clustered Semi-Asynchronous Federated Learning Framework. In IJCNN. IEEE, Shenzhen, China, 1–10.

**Questions:**

Q1. Are there experimental results on the datasets of FedWeIT under its setting?

Q2. How to select the correct concept model during inference/test as there are no data labels on the testset for Client Concept Matching?

Q3. Can we regard it as privacy risk to update and upload only selected concept models after local training?

---

> ### Author Response · Authors · 2023-11-19
>
> Thank you for appreciating our work on **the novel idea of learning a model set for clients to select in FL, the introduction of the two main challenges in FCL**. We address your comments as follows.
>
> **W1**. There are some more related works [1, 2] about clustered federated learning not included in this paper.
>
> **Response**:
> We added them in the updated version. Let us note that these two works are related only in the sense that they apply clustering in federated learning. They do not tackle any FCL challenges as our proposed approach.
>
>
> **W2**. The authors cite several related works about FCL but compare only one of them in experiments.
>
> **Response**:
> In addition to vanilla FL, our evaluation adapts two baselines. One baseline (FedWeIT) is the most cited FCL work published recently in a top machine learning conference. The other baseline is to apply a classic CL method (EWC) in FL. We believe they are representative of the state-of-the art.
>
>
> **W3**. This paper claims that CM can handle both class-incremental and task-incremental FCL but the experiments only include class-incremental settings.
>
> **Response**:
> In 5.1 experimental setup of the original submission, we mention that in task-incremental scenarios, the difference is that the clients do not have to perform concept matching every round, because they know the task IDs and the concept drift due to the transition of different tasks. For the same experimental setting, task-incremental training would achieve the same model performance with lower computation overhead at the clients.
>
> **Q1**. Are there experimental results on the datasets of FedWeIT under its setting?
>
> **Response**:
> We evaluate the proposed approach on the same data settings of FedWeIT. However, as we mention in 5.2 results  of the original submission, the learning settings of FedWeIT are on a very small scale and unrealistic (i.e. 5 clients per round, 5 classes to train per client, and non-overlapping classes over clients). In addition, the server in FedWeIT knows the class subset labels from the clients. Since our goal is to evaluate under realistic assumptions and on a larger scale, these learning settings are different from the original FedWeIT paper.
>
> **Q2**. How to select the correct concept model during inference/test as there are no data labels on the testset for Client Concept Matching?
>
> **Response**:
> In the updated paper, we added a clarification in 4.3 design discussion. For task-incremental training, during inference, the clients can pick the corresponding global concept model to perform prediction with their input data. For the class-incremental scenario, at inference, the clients can apply an ensemble model method to produce an output from all the global concept models.
>
>
> **Q3**. Can we regard it as privacy risk to update and upload only selected concept models after local training?
>
> **Response**:
> In the original submission, we explain in 4.3 design discussion that in terms of privacy, the CM framework is the same as vanilla FL (i.e., the clients send only their model weights to the server). In the updated paper, we further added the explanation that an adversarial server might seek to extract information from the client model using the same techniques from vanilla FL, which is beyond the scope of this paper.

---

> > ### Comment · Reviewer_anyU · 2023-11-21
> >
> > Response to W2:
> > FedWeIT adopts a dynamic model expansion scheme similar to CM in this paper. EWC adopts the regularization-based scheme. Could you add the replay-based method FedCIL as a comparison?
> >
> > Response to W3:
> > I agree with the author that task-incremental can be considered a slightly simpler task than class-incremental because of the task-ID signal. However, since there is no experimental verification, we cannot conclude that CM can still maintain superiority over the comparison methods in the task-incremental setting.
> >
> > Response to Q2 (task-incremental):
> > Since the experimental evaluation only relates to the class-incremental setting, can we conclude that the method described in Section 4.3 about the task-incremental setting is not supported by the experiments? Since concept model-ID and task-ID are bound by mapping in the task-incremental scenario, it seems unnecessary to perform concept matching during training. So there is no need to discuss the problem of task-incremental in this paper.
> >
> > Response to Q2 (class-incremental):
> > For the class-incremental setting, concept models are selected with concept matching for training. However, the ensemble scheme used for inference is not discussed in detail in the updated version. If it's just a naive ensemble mechanism that doesn't match the training phase, will the unrelated models interfere with the prediction?
> >
> > Response to Q3: I agree that the CM framework proposed in this paper employs a similar model uploading and aggregation scheme to vanilla FL. However, due to the maintenance of multiple global models, each client chooses different concept models to train and upload, which may lead to higher risks than vanilla FL, though it is not the main focus of this paper.

---

> > > ### Author Response · Authors · 2023-11-22
> > >
> > > Thank you for the new comments. We hope our previous responses clarify your other concerns in the original comments.
> > >
> > >
> > > **Comment**: I agree with the author that task-incremental can be considered a slightly simpler task than class-incremental because of the task-ID signal. However, since there is no experimental verification, we cannot conclude that CM can still maintain superiority over the comparison methods in the task-incremental setting.
> > >
> > > **Response**: We run additional experiments under the task-incremental setting. In the updated paper Appendix C.6 Figure 4, the learning curves of both the task-incremental and class-incremental settings exhibit smooth learning progress, and little difference in terms of model performance. The paper refers to this figure in 5.1 experimental setup when discussing the difference between task-incremental training and class-incremental training.
> > >
> > > **Comment (task-incremental)**: Since the experimental evaluation only relates to the class-incremental setting, can we conclude that the method described in Section 4.3 about the task-incremental setting is not supported by the experiments? Since concept model-ID and task-ID are bound by mapping in the task-incremental scenario, it seems unnecessary to perform concept matching during training. So there is no need to discuss the problem of task-incremental in this paper.
> > >
> > >
> > > **Response**: In the updated paper, we added the clarification in Section 4.3 that in both the task-incremental and class-incremental scenarios, the server does not know the concepts of clients data, and it shall perform clustering and concept matching every round. We rephrased “the clients do not have to perform concept matching every round” into “the clients only need to perform concept matching at the initial rounds”  for the task incremental-scenarios. In other words, the difference of CM adapted in the class-incremental and task-incremental is that the client in the task-incremental scenarios can use the mapping after it encounters all tasks. The other operations proposed in this study are all necessary for both task-incremental and class-incremental.
> > >
> > > **We agree with your other new comments**:  more baselines will strengthen the paper, different ensembling methods may make difference, and the privacy issue can be further explored for FCL distribution change in general, but ensembling methods or privacy are not the main focus of this paper.

---

### Official Review · Reviewer_Ustz · 2023-10-31

**Soundness:** 2 fair
**Presentation:** 2 fair
**Contribution:** 2 fair
**Rating:** 6
**Confidence:** 3

**Summary:**

The authors investigate a continual learning setting in a federated learning framework. They propose concept matching as a method to avoid the catastrophic forgetting problem in continual learning under the assumptions of a federated framework, i.e., that only model weights can be shared between clients and server. The concept matching algorithm:
-starts by sending a set of K concept-specific models to each client
-the client evaluates each model and fine-tunes the locally best-performing model
-the locally fine-tuned models are sent back to the server
-the server clusters the models into J clusters within which models are aggregated
-the server matches the J aggregated models with the initial K models to only update the matched ones
-the next round commences

**Strengths:**

The continual learning setting is relevant for many real-world applications. Investigating the interaction with restrictions from federated frameworks is valuable.

The authors provide some limited theoretical intuition for their algorithm and empirical evidence that it works.

**Weaknesses:**

The algorithm requires each client to evaluate each model at every round. How does this impact run time?

The experiments are based on a single "super" dataset which is of a relatively modest input scale.

The experiments do not seem to have been run multiple times with different random seeds.

**Questions:**

What happens if some concept models never get updated?

It would be helpful to have explicit definitions of concept and concept model in the introduction.

---

> ### Author Response · Authors · 2023-11-18
>
> Thank you for appreciating our work on **investigating CL in federated framework for real-world applications,  demonstrating the effectiveness of proposed approach through theoretical intuition and empirical evidence**. We address your comments as follows.
>
>
> **Weaknesses**:
>
> **1**. The algorithm requires each client to evaluate each model at every round. How does this impact run time?
>
> **Response**: We evaluate the client run time on a real IoT device in the original submission.  Due to the page limit, the details are in Appendix C.4, and the results are presented in Appendix C.6.  The total client end-to-end operation time in one round is 97.81 seconds, which is feasible for a real-world deployment. The average time for a client to evaluate models is 10% over the client training time,  and we believe the improvement in performance achieved by CM is worth this overhead cost.
>
> **2**. The experiments are based on a single "super" dataset which is of a relatively modest input scale.
>
> **Response**: Using the “super” dataset follows the most cited FCL work [1] published recently in a top machine learning conference. The dataset covers 183 image classes with 292250 images. We believe this scale is sufficient to evaluate the proposed approach.
> [1] Federated continual learning with weighted inter-client transfer, International Conference on Machine Learning, 12073--12086, 2021.
>
> **3**.The experiments do not seem to have been run multiple times with different random seeds.
>
> **Response**: Due to page limit, we mention in the appendix C.3 of the original submission that we test CM with different hyper-parameters, and only present the results with the hyper-parameters that lead to the best results. The hyper-parameters here include different random seeds. We moved this statement to the main paper in the updated version.
>
> **Questions**:
>
> **1**. What happens if some concept models never get updated?
>
> **Response**: Due to the effectiveness of the proposed approach, concept matching achieves up to 100% accuracy, and we do not observe the situation that some concept models never get updated. A possible situation that some concept models never get updated is when we intentionally start the system with more concept models than the number of concepts. Figure 6 in Appendix of the original submission illustrates this situation, the testing accuracy on each concept data is simply not affected by the extra models in the system.
>
> **2**. It would be helpful to have explicit definitions of concept and concept model in the introduction.
>
> **Response**:  In the updated submission, we moved the explicit definitions of concept and concept model from section 2 overview to section 1 introduction.

---

### Official Review · Reviewer_HPc6 · 2023-11-04

**Soundness:** 2 fair
**Presentation:** 2 fair
**Contribution:** 1 poor
**Rating:** 3
**Confidence:** 4

**Summary:**

This paper proposed a Concept Matching (CM) method for Federated Continual Learning (FCL). The main innovation of this article lies in maintaining global concept models for matching on the server . At training step, each client selectively updates the weights of its corresponding model. These updated weights are then clustered on the server side to differentiate between different tasks and reassign them to their respective matches. From a technical perspective, the approach presented in this paper is straightforward. Experimental results indicate that the proposed method outperforms the benchmark.

**Strengths:**

1. Federated Continual Learning is a genuine real-world problem that exists in practical scenarios.
2. The experimental results clearly demonstrate the effectiveness of the proposed method within the experimental setting provided by the authors.
3. The algorithm framework proposed in the paper is presented in a concise and easily understandable manner.

**Weaknesses:**

1.The problem background is unreasonable and does not align with practical needs. The requirement for each client to train with different datasets each time is not realistic in real-world scenarios.
2.The method lacks innovation and can be seen as a mere patchwork of existing client aggregation approaches.
3.The experimental comparisons lack persuasiveness as the baselines should be diverse in their configurations.

**Questions:**

1.In the paper, there is a lack of explanation regarding the calculation of distlist[k] in line 9 of Algorithm 1. The author should provide further clarification on this point to enhance the understanding of the algorithm.

2.The model's implicit assumption that the number of concepts is smaller than the number of global concept models introduces a limitation. When there is a large number of concepts, it becomes evident that the algorithm's maintained models may encounter difficulties in effectively handling this situation. This limitation is a result of the algorithm's design.

3.The overall design of the model lacks novelty, as there are many similar methods available. The author simply transfers existing methods from other domains to the problem of federated continual learning without introducing any substantial innovation.

4.The experiments should be more comprehensive and diversified. It is recommended to include baselines that cover scenarios where all the data is combined or where all the data is separated. This will ensure a fair comparison and prevent experiments from being solely designed to favor the proposed method.

---

> ### Author Response · Authors · 2023-11-20
>
> Thank you for appreciating our work on **addressing a genuine real-world problem, demonstrating the effectiveness of the proposed method with the experiment, and presenting the framework in a concise and easily understandable manner**. We address your comments as follows.
>
> **W1**.The problem background is unreasonable and does not align with practical needs. The requirement for each client to train with different datasets each time is not realistic in real-world scenarios.
>
> **Response**: In the introduction and overview, we state the motivation background for our problem as FCL for IoT and mobile devices, where it is difficult to train with the entire dataset on-device at every round due to resource constraints, and the data not only accumulate over time, but also change their distributions.  We provide human activity recognition (HAR) as an example, and its distribution change can be due to the subsets of activities, the locations of the activities, or the health status of the user.
> It seems the reviewer only focused on the scenario in Figure 1 where each client trains with different image classes each time. Nevertheless, we added another example of such a real-world scenario in the updated paper. For the photos a client takes in daily life, the concepts can be different types of the place of interest visited (e.g., museum vs. national park), or different types of meals (i.e. breakfast, lunch, dinner).
>
>
> **W2**.The method lacks innovation and can be seen as a mere patchwork of existing client aggregation approaches.
>
> **Q3**. The overall design of the model lacks novelty, as there are many similar methods available. The author simply transfers existing methods from other domains to the problem of federated continual learning without introducing any substantial innovation.
>
> **Response**: The novelty was further clarified in the updated paper. There are two main novelties in the design. The first novelty is the clustering-based framework to alleviate catastrophic forgetting and interference among clients, and consequently achieve good model performance in FCL. The second novelty is our effective and theoretically grounded concept matching algorithms. Because the server does not know which cluster model is responsible for which concept, clustering alone will not solve FCL.
>
>
> **W3**.The experimental comparisons lack persuasiveness as the baselines should be diverse in their configurations.
>
> **Q4**.The experiments should be more comprehensive and diversified. It is recommended to include baselines that cover scenarios where all the data is combined or where all the data is separated. This will ensure a fair comparison and prevent experiments from being solely designed to favor the proposed method.
>
> **Response**:
> - In the updated submission, we added the comparison results between vanilla FL and CM where all the data is separated (i.e., treating each dataset as a concept without mixing or splitting the datasets). Similar to the orignal settings, Figure 3 in Appendix C.5 illustrates the smooth learning curves for CM over vanilla FL for all concepts.Compared with vanilla FL, the weighted average accuracy over the number of samples for all the concepts is improved from 85.5% to 88.0% as well.
> - We do not test the scenario with  all the data combined, because it is not a CL setting.
> - In the original submission, we explain in Appendix C.1 that we split FaceScrub into two concepts to verify whether CM can differentiate them successfully. We mix the three MNIST datasets together to make it more difficult, because MNIST datasets are easy to learn. The purpose is to stress-test CM rather than favoring the proposed method.
>
> **Q1**.In the paper, there is a lack of explanation regarding the calculation of distlist[k] in line 9 of Algorithm 1. The author should provide further clarification on this point to enhance the understanding of the algorithm.
>
> **Response**: We fixed the typo, by replacing distlist[k] in line 9 as distRecord[k] (defined in line 3).
>
> **Q2**.The model's implicit assumption that the number of concepts is smaller than the number of global concept models introduces a limitation. When there is a large number of concepts, it becomes evident that the algorithm's maintained models may encounter difficulties in effectively handling this situation. This limitation is a result of the algorithm's design.
>
> **Response**: The paper does not assume the number of concepts is smaller than the number of global concept models. On the contrary, our evaluation shows CM has good resilience when configured with numbers of concept models that are different (either smaller or larger) from the number of concepts. The results are in the Appendix C.6 due to the page limit.  Even when the number of concepts is larger than the number of global concept models,  the average model accuracy achieves 89.5% and 88.9% respectively, and beats vanilla FL (86.7%).

---

### Official Review · Reviewer_2nMo · 2023-11-07

**Soundness:** 2 fair
**Presentation:** 2 fair
**Contribution:** 2 fair
**Rating:** 3
**Confidence:** 4

**Summary:**

In order to overcome the two issues of catastrophic forgetting and the potential interference among clients in Federated Continual Learning (FCL), the authors proposed a clustering-based framework, called Concept Matching. At each time step, the CM framework first assigns a concept model for each client as initialization in the fine-tuning process. Then the client models trained with local data, are clustered into some groups, which are used to update those concept models via the designed server concept matching approach in Algorithm 1. Finally, the updated concept models are for the next time step.

**Strengths:**

The proposed concept matching framework has used different clustering, aggregation, and concept matching algorithms, which can effectively improve performance compared with the state-of-the-art systems in Federated Continual Learning.

**Weaknesses:**

1. In terms of equations, many equations are not written clearly and normatively. Specifically, Eq.(2) is wrong where k* has missed n in Eq.(2), and the argmin operation of the loss function is not presented. Aggregate function in Eq.(4) is not expressed. \Theta and \omega are not unified in \Theta^t={\omega_1,\omega_2,...,\omega_J}.
2. In terms of the writing, this submission has not been written well. What is the meaning of the concept in the image experiment of this paper? This abstract vocabulary has not been explained clearly. More reasons for experiments should be analyzed. Many equations are not clear. In Algorithm 1, the \omega_j and \omega_k are repetitive in the inner and outer loop.
3. In terms of experiments, the experiments are also insufficient. It is very important to discuss the number of concept models and clustered groups, as well as their relationship from the perspective of theory or experiment, which can help readers understand the importance of clustering for FCL. The authors are suggested to provide more analysis and reasons about all experiments, such as performance differences based on various distance functions, clustering methods, etc.
4. The authors claim that two issues in Federated Continual Learning (FCL), catastrophic forgetting and interference among clients, can be greatly diminished. It will be better to prove that claim through conducting some experiments.
5. The novelty is limited, as there are some Federated Learning (FL) works adopting the clustering method. however, the authors apply the clustering method to FCL at a single time step, which does not reflect the unique design of CL.

**Questions:**

Shown in the above Weaknesses.

---

> ### Author Response · Authors · 2023-11-19
>
> Thank you for appreciating our work on **the effective performance improvement with different clustering and aggregation algorithms**. We address your comments as follows.
>
> **Weaknesses**
>
> **1**.In terms of equations,many equations are not written clearly and normatively.Specifically,Eq.(2) is wrong where k* has missed n in Eq.(2),and the argmin operation of the loss function is not presented.Aggregate function in Eq.(4) is not expressed.\Theta and \omega are not unified in \Theta^t={\omega_1,\omega_2,...,\omega_J}.
>
> **Response**
> - Eq.(2) is a client operation, k* is a client local value unrevealed to the server. Thus, n is omitted in the original submission.  Due to the confusion, we added n in the updated paper.
> - We do not find any equation with loss function in the original submission where the argmin operation is absent. They are all presented when expressing the optimization of the loss function, including eq(1) and eq(6). It will be helpful if the reviewer could indicate the specific equation in the paper where the argmin operation of the loss function is not presented.
> - From the abstract and throughout the paper, we state the CM framework is flexible to use any aggregation function. Thus, we did not specifically express any aggregation function.
> -  \Theta (aggregated cluster model set) is used to differentiate the global model set \Omega, even though its elements are also models. Thus, we used a different symbol.
>
>
> **2**. In terms of the writing,this submission has not been written well.What is the meaning of the concept in the image experiment of this paper?This abstract vocabulary has not been explained clearly.More reasons for experiments should be analyzed. Many equations are not clear.In Algorithm 1,the \omega_j and \omega_k are repetitive in the inner and outer loop.
>
> **Response**
> - In the original submission, we define the image concepts as different sets of classes from the first time we mention image in section2 overview.
> - We updated the abstract. We would greatly appreciate it if the reviewer could provide more specific details regarding the aspects in which the abstract is unclear.
> - The first paragraph in evaluation of the original submission specifies the main goals of the experiments as well as minor goals for Appendix. Our experiments are designed to target these goals.
> - \omega_j and \omega_k are elements in different sets. We fixed the minor issue due to the confusion.
>
> **3**. In terms of experiments,the experiments are also insufficient.It is very important to discuss the number of concept models and clustered groups,as well as their relationship from the perspective of theory or experiment,which can help readers understand the importance of clustering for FCL.The authors are suggested to provide more analysis and reasons about all experiments,such as performance differences based on various distance functions,clustering methods,etc.
>
> **Response**
>
> All these experiments the reviewer considered insufficient are already part of the original submission.
> - The number of concept models, clustered groups and their relationship are evaluated and discussed as **Performance of clustering algorithms** and **Resilience to number of concepts different from the ground truth** in the paper.
> - We illustrate CM performance based on various distance functions, clustering methods in Table 2 and Table 3. The difference is negligible in terms of model accuracy, and we conclude in the paper that CM provides the flexibility to use different clustering algorithms and distance metrics, as it performs well with all of them. As neural networks grow in size and complexity, the curse of dimensionality may manifest itself and requires advanced clustering algorithms and distance metrics.
>
>
> **4**.The authors claim that two issues in FCL,catastrophic forgetting and interference among clients,can be greatly diminished.It will be better to prove that claim through conducting some experiments.
>
> **Response**
>
> Figure 2 in the original submission illustrates the negative effect of catastrophic forgetting and interference among clients on the learning curve and the model performance in vanilla FL, and CM can effectively smoothen the learning curve and improve the model performance. We will investigate how to further quantify catastrophic forgetting and interference among clients in the next version.
>
>
> **5**.The novelty is limited,as there are some FL works adopting the clustering method.however,the authors apply the clustering method to FCL at a single time step,which does not reflect the unique design of CL.
>
> **Response**
>
> The reviewer misunderstood the proposed approach. Throughout the paper, we indicate clustering is applied every FCL training round. Clustering alone will not solve FCL, because the server does not know which cluster model is responsible for which concept. Our additional novelty lies in our effective and theoretically grounded concept matching algorithms.

---

### Meta-Review · Area_Chair_Q6fi · 2023-12-06

**Metareview:**

This paper proposes a Concept Matching (CM) method for Federated Continual Learning (FCL). The primary innovation in this article lies in maintaining global concept models for matching on the server. During the training step, each client selectively updates the weights of its corresponding model. These updated weights are then clustered on the server side to differentiate between different tasks and reassign them to their respective matches. As acknowledged by most reviewers, the approach presented in this paper is straightforward.
Before rebuttal, the experiments are insufficient, and the motivation is not clearly explained. The reviewers did not show any support to this paper. The authors are encouraged to provide more in-depth analysis and reasons regarding all experiments in other future venue.

**Justification For Why Not Higher Score:**

N/A

**Justification For Why Not Lower Score:**

N/A

---

### Decision · Program_Chairs · 2024-01-16

Reject